# Resmax: An Alternative Soft-Greedy Operator for Reinforcement Learning

**Erfan Miahi**                                                            *miahi@ualberta.ca*
*Department of Computing Science*
*University of Alberta*

**Revan MacQueen**                                                        *revan@ualberta.ca*
*Department of Computing Science*
*University of Alberta/Amii*

**Alex Ayoub**                                                            *aayoub@ualberta.ca*
*Department of Computing Science*
*University of Alberta*

**Abbas Masoumzadeh**                                                 *masoumza@ualberta.ca*
*University of Alberta*

**Martha White**                                                        *whitem@ualberta.ca*
*Department of Computing Science*
*University of Alberta*

**Reviewed on OpenReview:** *https://openreview.net/forum?id=wzzrs5QH5k&noteId=9pzVlZ7sC9*

## Abstract

Soft-greedy operators, namely $\varepsilon$-greedy and softmax, remain a common choice to induce a basic level of exploration for action-value methods in reinforcement learning. These operators, however, have a few critical limitations. In this work, we investigate a simple soft-greedy operator, which we call resmax, that takes actions proportionally to their max action gap: the residual to the estimated maximal value. It is simple to use and ensures coverage of the state-space like $\varepsilon$-greedy, but focuses exploration more on potentially promising actions like softmax. Further, it does not concentrate probability as quickly as softmax, and so better avoids overemphasizing sub-optimal actions that appear high-valued during learning. Additionally, we prove it is a non-expansion for any fixed exploration hyperparameter, unlike the softmax policy which requires a state-action specific temperature to obtain a non-expansion (called mellowmax). We empirically validate that resmax is comparable to or outperforms $\varepsilon$-greedy and softmax across a variety of environments in tabular and deep RL.

## 1 Introduction

Many value-based methods in reinforcement learning rely on soft greedy operators, such as *softmax* (Luce, 1959) and $\varepsilon$-*greedy* (Watkins, 1989). These operators play two roles: for soft-greedification within the Bellman update when bootstrapping off values in the next state (target policy) and to encourage some amount of exploration (in the behavior policy). Soft-greedification within the Bellman update facilitates learning a stochastic soft-optimal policy and can help mitigate the overestimate bias in Q-learning (Song et al., 2019). This changes the target policy being learned. For the second role, regardless of whether a Q-learning update is used or a soft-greedy update, the soft-greedy operator can be used to take actions. Such an approach is actually complementary to directed exploration approaches, like those that learn optimistic values, because it can provide a small amount of additional exploration and so robustness to estimation error in the optimistic

values. And for some environments, where only a small amount of exploration is needed, these soft-greedy operators for the behavior provide a sufficient level of exploration.

The two most common soft-greedification operators—$\varepsilon$-greedy and softmax—have several limitations. $\varepsilon$-greedy explores in an undirected way. Regardless of the agent's estimates of the value of an action, the agent's exploration step is uniformly random. Softmax—a Boltzmann policy on the action-values—is more directed, in that it samples proportionally to the exponentials of the value of the actions. However, softmax is notoriously difficult to tune and suffers from concentrating too quickly due to the use of this exponential. This concentration overemphasizes actions that appear high-valued under current estimates but are actually suboptimal. This *overemphasis* issue can cause softmax to settle on a suboptimal policy, as we reaffirm in our experiments. Further, the softmax is not sound when used within the Bellman update: it does not guarantee that the Bellman update is a contraction (Asadi & Littman, 2017), which is needed to ensure convergence under Sarsa or dynamic programming updates (Littman & Szepesvári, 1996). The greedy or $\varepsilon$-greedy operators, on the other hand, are both *non-expansions* and so ensure convergence: when used inside the Bellman update, with a discount less than 1 or under proper policies, the Bellman update is a contraction.

There has been work improving on these soft-greedy operators for value-based techniques. Several works have addressed the non-expansion issue for softmax (Asadi & Littman, 2017; Cesa-Bianchi et al., 2017; Pan et al.). One of these works proposes a soft-greedy operator called *mellowmax* (Asadi & Littman, 2017) that is guaranteed to be a non-expansion when it is used within the Bellman update. The operator involves optimizing the exploration parameter in softmax per state and has been shown to improve stability when it is used for exploration (Kim et al., 2019). However, mellowmax is not a simple heuristic to use, as it requires solving a root finding problem to compute the policy for decision making. To solve this problem they used Brent's method, which is computationally complex (Wilkins & Gu, 2013). An approach called value-difference exploration (Tokic, 2010; Tokic & Palm, 2011) adapts the exploration parameter $\varepsilon$ and the temperature for softmax over time, using the difference in the softmax of the values before and after learning. A later empirical study, however, highlighted that this approach does not perform consistently (Gimelfarb et al., 2020).

In this work, we consider a new soft-greedy operator, which we call resmax, based on a probability matching scheme originally developed in the contextual bandit setting (Abe & Long, 1999; Foster & Rakhlin, 2020). This technique is similar to softmax in the sense that it assigns distinct probabilities to actions based on the estimated action-values. However, unlike softmax, the probability for taking each action is determined using its max action gap: the difference between the approximated value of the greedy action and the given action. The policy is inversely proportional to this max action gap, and avoids the use of the exponential that causes softmax to overemphasize actions. We theoretically show that it ensures a minimal probability on each action, regardless of the action-values, ensuring all actions are explored. We prove that it is a non-expansion, and so combines well with generalized value iteration algorithms. We additionally provide an empirical study, across a variety of hard and easy exploration problems, with tabular and deep function approximation. We find that, in tabular experiments, resmax outperforms both softmax and $\varepsilon$–greedy–especially when softmax suffers from overestimation– and performs similarly to mellowmax, but with dramatically lower run time. In the deep RL experiments, resmax is comparable to the baselines.

**Remark:** It is worth noting that parallel work in the policy gradient literature examines different policy parameterizations, that resemble some of these operators used for value-based methods. It has been noted that the softmax policy parameterization can concentrate too quickly (Mei et al., 2020), motivating the introduction of another parameterization called the escort transform. Other work, particularly those analyzing convergence properties, examines categorical policy parameterizations (Zhan et al., 2021). This policy gradient work does not directly apply here, because we do not learn a parameterized policy. Instead, we use these operators to obtain a policy directly from learned action-values. Policy gradient methods often learn *action preferences*, which are not the same as action-values, resulting in notably different properties. For example, action preferences can concentrate very quickly, making entropy regularization more critical in policy gradient methods. Further, the non-expansion requirement is unnecessary for policy gradient methods; in fact, we highlight that an extension of the escort transform to this value-based setting does not have the non-expansion property.

## 2   Background

We model the environment as a discounted Markov Decision Process (MDP) $(\mathcal{S}, \mathcal{A}, \mathcal{R}, P, \gamma)$ where $\mathcal{S}$ is the set of states; $\mathcal{A}$ the set of actions; $\mathcal{R}$ the set of possible rewards; $\gamma \in [0, 1]$ the discount factor; and $P : \mathcal{S} \times R \times \mathcal{S} \times \mathcal{A} \to [0, 1]$ the dynamics function. In a given state $s$, the agent takes action $a$ and transitions to state $s'$ and receives reward $r$ according to probability $P(s', r|s, a)$.

The agent's goal is to learn a policy $\pi : \mathcal{S} \times \mathcal{A} \to [0, 1]$ that maximizes its discounted cumulative reward. The action-value function for each state-action pair under the policy is

$$q_\pi(s, a) \doteq \mathbb{E}_\pi \left[ \sum_{k=0}^\infty \gamma^k R_{t+k+1} | S_t = s, A_t = a \right] \tag{1}$$

The agent attempts to approximate $q_{\pi^*}$, the action-value function for the optimal policy $\pi^*$, with an approximate action-value function $q$ that is parameterized by $\theta$. Many reinforcement learning algorithms compute this approximation iteratively, using either Q-learning or Expected Sarsa. Both methods use an update to the parameter vector $\theta$ of the form

$$\theta_{t+1} \leftarrow \theta_t + \alpha \delta_t \nabla q(s, a, \theta_t)$$

but with different TD errors $\delta_t$. Q-learning uses $\delta_t = r + \gamma \max_{a'} q(s', a', \theta_t) - q(s, a, \theta_t)$. Expected Sarsa uses $\delta_t = r + \gamma \mathbb{E}_{a' \sim \pi(\cdot|s')}[q(s', a', \theta_t)] - q(s, a, \theta_t)$ where the target policy $\pi$ is typically a soft-greedy policy. When $\theta$ has an entry for each $s, a$, pair, we call this the *tabular setting*. Going forward, we will use $q(s, a)$ in our descriptions of operators and policies; however, one can easily substitute it for $q(s, a, \theta)$ in non-tabular settings.

To learn these action-values, the agent needs to explore. One simple strategy to promote exploratory behavior is to use soft-greedy policies, either just for the behavior (as in Q-learning) or for both the behavior and target (as in Expected Sarsa). Two common soft-greedy policies are the Boltzmann softmax policy and $\varepsilon$-greedy, defined as

$$\pi_{\mathrm{sm}}(a \mid s) \doteq \frac{e^{q(s,a)\tau^{-1}}}{\sum\limits_{a' \in \mathcal{A}} e^{q(s,a')\tau^{-1}}} \tag{2}$$

$$\pi_{\varepsilon\mathrm{g}}(a \mid s) \doteq \begin{cases} 1 - \varepsilon + \frac{\varepsilon}{|\mathcal{A}|} & \text{if } a = \mathrm{argmax}_a q(s, a) \\ \frac{\varepsilon}{|\mathcal{A}|} & \text{otherwise} \end{cases} \tag{3}$$

with an exploration parameter $\tau > 0$ and $\epsilon \in [0, 1]$. With lower values of $\tau$, softmax is greedier; with higher values of $\tau$, softmax action selection becomes more equiprobable. Similarly, for $\epsilon = 0$, the $\varepsilon$-greedy policy is perfectly greedy, becoming increasing random with higher $\varepsilon$; at $\varepsilon = 1.0$, the policy becomes uniform random with equiprobable action selection. Thus, these methods share two essential characteristics: randomness in action selection to guarantee the coverage of the whole state space and a hyperparameter to vary the degree to which the greedy action is chosen.

Though widely used, softmax and $\varepsilon$-greedy suffer from several flaws. One of the major problems of $\varepsilon$-greedy is that it ignores the estimates of action-values and assigns a uniform probability to each non-greedy action. This undirected exploration results in wasting time taking actions that the agent might already know are vastly suboptimal. Further, to explore specific parts of the environment, it might need to chain a sequence of random actions, which might be very low probability. Consider, for instance, the RiverSwim environment (Strehl & Littman, 2008), where an agent receives a minor positive reward if it takes the left action in the initial state, but a much larger reward at the far right of the environment that is harder to reach. To receive this high reward, the agent might need to take several exploratory actions in a row, and so typically gets stuck learning to go left.

Softmax is more directed, in that it assigns the probability of each action corresponding to its action-value. However, if the temperature is not set carefully, softmax will assign an excessively disproportionate probability

to the greedy action because this probability is based on exponentiating action values. Consequently, softmax often overemphasizes actions that currently have high value estimates, at the expense of exploring other actions. We provide an illustrative example in the next section, explaining this issue further.

Furthermore, when using the softmax operator in the Expected Sarsa update, this operation is not guaranteed to be a non-expansion (Littman, 1996; Littman & Szepesvári, 1996). The softmax operator inside the Bellman update, is $\text{sm}(q(s', \cdot), \tau) = \mathbb{E}_{a' \sim \pi_{\text{sm}}(\cdot|s')}[q(s', a')] = \sum_{a'} \pi_{\text{sm}}(a'|s')q(s', a')$. Generalized Value Iteration (GVI) algorithms, such as Expected Sarsa, require this operator to be a non-expansion, to guarantee converge to a *unique* fixed point (Littman & Szepesvári, 1996; Asadi & Littman, 2017). Without the non-expansion property, an operator may converge to multiple fixed points, even in the tabular setting; softmax does not have this property.

The mellowmax operator (Asadi & Littman, 2017)

$$\text{mm}_\omega(\mathbf{x}) \doteq \omega \log\left(\frac{1}{n}\sum_{i=i}^{n} e^{x_i \omega^{-1}}\right) \tag{4}$$

was designed to fix this non-expansion issue[1]. Mellowmax behaves as a quasi-arithmetic mean, with a parameter $\omega$ controlling how much mellowmax behaves like an average or a max. The corresponding mellowmax policy $\pi_{\text{mm}}$ can be found by solving a convex optimization. First, using a solver[2], one obtains a value $\beta$ such that:

$$\sum_{a \in \mathcal{A}} e^{\beta q(s,a) - \beta \text{mm}_\omega(q(s,\cdot))} q(s,a) - \text{mm}_\omega(q(s,\cdot)) = 0 \tag{5}$$

This $\beta$ is a state-specific softmax temperature to provide the resulting (maximum entropy) mellowmax policy:

$$\pi_{\text{mm}}(a|s) \doteq \frac{e^{q(s,a)\beta}}{\sum_{a' \in \mathcal{A}} e^{q(s,a')\beta}} \tag{6}$$

Since $\beta$ needs to be computed at each state, mellowmax needs considerably more compute than softmax, resmax or $\varepsilon$-greedy. The need for root finding also increases the complexity of implementing mellowmax as a soft-greedy operator; in our implementation, we had to correct for numerical instability and iteratively increase the bounds of the root finder to make sure a $\beta$ always exists without having bounds set too wide (and thus affecting runtime).

## 3 The Resmax Operator

Our goal is to obtain the benefits of all three of these operators: (a) use the information provided by action-values (like softmax and mellowmax), (b) avoid the overemphasis of softmax (like $\varepsilon$-greedy) and (c) be easy and computationally efficient to use (unlike mellowmax). For this purpose, we propose resmax. Let $q_{\max}(s) \doteq \max_a q(s,a)$ be the value of any greedy action in state $s$, and $G(s) = \{b \in \mathcal{A} \mid q(s,b) = q_{\max}\}$ be the set of greedy actions. The resmax policy $\pi_{\text{rm}}$ is defined as

$$a \notin G(s): \quad \pi_{\text{rm}}(a \mid s) \doteq \frac{1}{|\mathcal{A}| + \eta^{-1}(q_{\max} - q(s,a))} \tag{7}$$

$$b \in G(s): \quad \pi_{\text{rm}}(b \mid s) \doteq \frac{1}{|G(s)|}\left(1 - \sum_{a \notin G(s)} \pi_{\text{rm}}(a \mid s)\right)$$

where the exploration parameter $\eta > 0$ can be thought of as the *exploration pressure*. Larger $\eta$ values push towards exploration whereas low values result in more exploitation— are greedier. When $\eta \to \infty$ the policy is uniformly random and when $\eta \to 0$ the policy is greedy with respect to $q(s,b)$.

---

[1] We write mellowmax with $\omega \doteq 1/\omega$ in Asadi & Littman (2017), so that as $\omega$ increases, mellowmax behaves more greedily— similarly to $\varepsilon$ in $\varepsilon$-greedy and $\tau$ in softmax.

[2] We use brentq, provided in scipy (Virtanen et al., 2020).

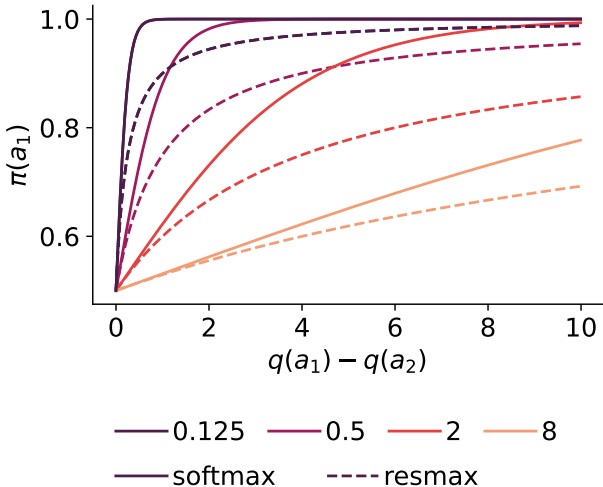

Figure 1: Contrasting policies produced using the softmax and resmax, for different exploration parameters, with $q(a_2) = -5$ and $q(a_1) \in [-5, 5]$. As the max action gap increases, softmax much more quickly concentrates probability on action $a_1$ than resmax.

In words, the resmax policy first assigns probability to every action inversely proportionally to it's action-values: $1/(|\mathcal{A}| + \eta^{-1}(q_{\max} - q(s,a)))$ is biggest when $q(s,a) = q_{\max}$, equaling $1/(|\mathcal{A}|)$, with less probability assigned to an action as $q(s,a)$ gets further from $q_{\max}$. Then the remaining probability is evenly distributed amongst the maximal (greedy) actions. This mimics $\varepsilon$-greedy, where all actions are assigned a small probability $\varepsilon/|A|$ and remaining probability distributed among maximal actions. Resmax, however, assigns small probabilities for each non-greedy action *proportional to the action-values*, rather than uniformly.

Consider two extreme cases for intuition. If all actions are equal, then they are all assigned $1/|\mathcal{A}|$. If one action $b$ is maximal, then all others actions are assigned some probability smaller than $1/|\mathcal{A}|$, and the remaining probability assigned to $b$ will be higher than $1/|\mathcal{A}|$. If the action gap is big, $q(s,b) >> q(s,a)$ and $\eta = 1$, for example, then $\pi_{\mathrm{rm}}(b|s)$ is almost 1 and the probability on the remaining actions is near zero.

This operator is efficient to compute and easy to use for exploration or soft-greedification. To use it with Q-learning or DQN, the agent simply samples actions from the resmax policy in Equation 7. To use it within the Bellman update, for soft-greedification, we can simply use the corresponding resmax operator

$$\mathrm{rm}(q(s,\cdot), \eta) \doteq \sum_{a \in \mathcal{A}} \pi_{\mathrm{rm}}(a \mid s) q(s,a) \tag{8}$$
$$= \sum_{a \in \mathcal{A}} \frac{q(s,a)}{|\mathcal{A}| + \eta^{-1}(q_{\max} - q(s,a))} + q_{\max}\left(1 - \sum_{a \in \mathcal{A}} \frac{1}{|\mathcal{A}| + \eta^{-1}(q_{\max} - q(s,a))}\right)$$

As $\eta$ goes to 0, this operator becomes the max operator (proven in Appendix A).

To better understand resmax, we contrast it to softmax in Figure 1. We particularly highlight that softmax overemphasizes actions with high approximates values, in comparison to resmax. We consider a setting where there are two actions $a_1$ and $a_2$, and visualize the policies for a state $\hat{s}$ for different max action gaps between the approximate values, $q(a_1) - q(a_2)$, where $q(a_i) \doteq q(\hat{s}, a_i)$. We set $q(a_2) = -5$ and then vary $q(a_1) \in [-5, 5]$. For most choices of the temperature $\tau$, if $q(a_1)$ is sufficiently larger than $q(a_2)$, then the outputs of the softmax policy for $a_1$ will be near 1. Thus, it may take a very large number of steps to finally choose action $a_2$ and explore this other action. Under resmax, the probability assigned to the greedy action is much less skewed towards 1, particularly for smaller gaps between the values in the actions.

In the next two sections, we motivate that resmax satisfies the other two requirements, in addition to ease-of-use: (1) it avoids the overemphasis in softmax and encourages exploration and (2) is a non-expansion.

## 4 Resmax Encourages Exploration

In this section, we first confirm that resmax is guaranteed to provide sufficient exploration, by maintaining non-neglible probability on all actions. Then we investigate resmax in two hard exploration problems, with misleading rewards, in comparison to softmax and mellowmax.

### 4.1 State-Action Space Coverage

Exploration strategies should satisfy certain fundamental properties to make sure that algorithms such as Q-learning and Sarsa converge to the optimal value (Singh et al., 2000; Watkins & Dayan, 1992). A key property is that each state-action pair should be visited infinitely many times during continual learning. To show that resmax satisfies this property, we prove that the probability of taking all of the actions will be higher than zero during learning for any bounded action-values. This result is straightforward to show, but needed for completeness to ensure we do not lose this useful property of $\varepsilon$-greedy and softmax.

*Property* 4.1. Assume there exists $q_{\mathrm{bound}} > 0$ such that $\forall s, a, |q(s, a)| \leq q_{\mathrm{bound}}$. The probability of taking any non-greedy action $a$ and a greedy action $b$ satisfies

$$0 < \frac{1}{|\mathcal{A}| + 2\eta^{-1} q_{\mathrm{bound}}} \leq \pi_{\mathrm{rm}}(a \mid s) < \frac{1}{|\mathcal{A}|} \leq \pi_{\mathrm{rm}}(b \mid s) < 1$$

*Proof.* First, we determine the upper-bound and lower-bound for non-greedy actions $a$. By analyzing Equation 7, it is clear that its lowest-value will be obtained only when the difference between $q(s, b)$ and $q(s, a)$ is at its highest. We therefore get the following lower bound on $\pi_{\mathrm{rm}}(a \mid s)$

$$\begin{aligned}
\pi_{\mathrm{rm}}(a \mid s) &= \frac{1}{|\mathcal{A}| + \eta^{-1}(q(s, b) - q(s, a))} \\
&\geq \frac{1}{|\mathcal{A}| + \eta^{-1}\left(q_{\mathrm{bound}} - (-q_{\mathrm{bound}})\right)} \\
&= \frac{1}{|\mathcal{A}| + 2\eta^{-1} q_{\mathrm{bound}}} > 0
\end{aligned}$$

Furthermore, $\pi_{\mathrm{rm}}(a \mid s)$ is bounded above by $1/|\mathcal{A}|$, because $q(s, b) - q(s, a) > 0$ in the denominator. The probability of greedy actions $b$ are lowest when all actions are greedy, having probability $1/|\mathcal{A}|$. $\qquad\square$

### 4.2 An Illustrative Example of Overemphasis

A serious issue with using softmax for exploration comes from the fact that it uses exponents within its formulation, which can assign an overly disproportionate probability to current greedy action under the approximation action values. We demonstrate the severity of the softmax overemphasis problem in the HardSquare MDP (Figure 2a). The agent starts in the state $s_1$ and $s_2$ with an equal probability. The agent can stay in the start states for a reward of $10^4$, or move to $s_3$ and $s_4$, where it can receive a larger reward of $2 \times 10^4$. In expectation, $s_3$ and $s_4$ are better, but the high reward in the initial states $s_1$ and $s_2$ can mislead the agent. For the experiment, we use tabular Q-learning. We initialize $q(s, a) \leftarrow 0$, $\forall (s, a) \in \mathcal{S} \times \mathcal{A}$. The results, depicted in Figure 2b, are obtained by averaging over 30 runs and selecting the best performance across three different step sizes.

We can see that the softmax policy stays in the initial states $s_1$ and $s_2$ due to its tendency to overemphasize the actions that stay in the start states. On the contrary, resmax can escape from the initial states and successfully explore towards the optimal solution. We include mellowmax in our results for Hardsquare and note that it performs worse than resmax for best choices of hyperparameters. Due to the optimization procedure inherent in mellowmax, its runtime is on average $28\times$ longer than resmax.

### 4.3 More Experiments in a Classic Hard Exploration Environment

We further investigate the exploration properties of these operators in a more well-known environment typically used to test exploration, called RiverSwim (Strehl & Littman, 2008). This environment consists

of 6 states arranged in line, of which the agent starts in the leftmost. This state has a small reward but the rightmost state may give a comparatively large reward. In order to reach this state the agent must traverse the MDP from left to right, fighting against a current which causes moving right to often fail. We use RiverSwim with a fixed horizon, meaning that after 20 actions the agent will be returned to the start state, in effect making the environment more difficult. See Figure 3 for the full specification. All algorithms are run for 800,000 time steps.

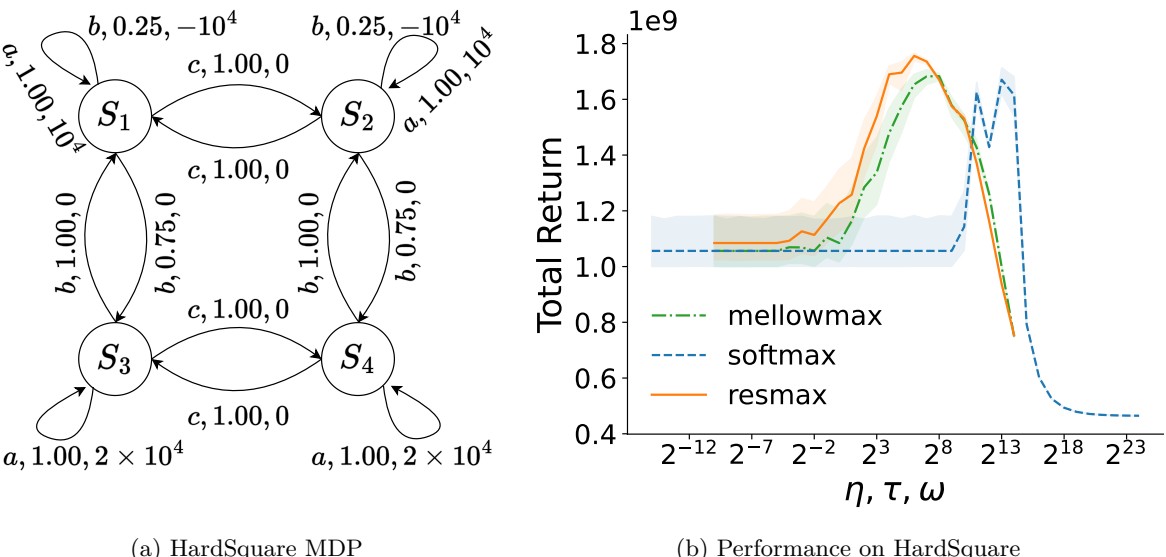

(a) HardSquare MDP

(b) Performance on HardSquare

Figure 2: (a) The HardSquare MDP: a simple MDP with four states, three actions and $\gamma = 0.95$. Each edge shows an stochastic transition labeled by action, probability, and reward respectively. The agent starts in states $s_1$ and $s_2$ with equal probability. (b) Performance of resmax, mellowmax and softmax on HardSquare. The means and standard errors are shown for 30 runs for each parameter setting. The x-axis is the hyper-parameter choices for each operator. Softmax gets stuck in states $s_1$ and $s_2$, whereas resmax can escape from the initial states and successfully explore towards higher value states. Results are shown for $\eta, \omega \in \{2^{-10}, 2^{-9}, ..., 2^{15}\}$ and $\tau \in \{2^{-15}, 2^{-14}, ..., 2^{25}\}$.

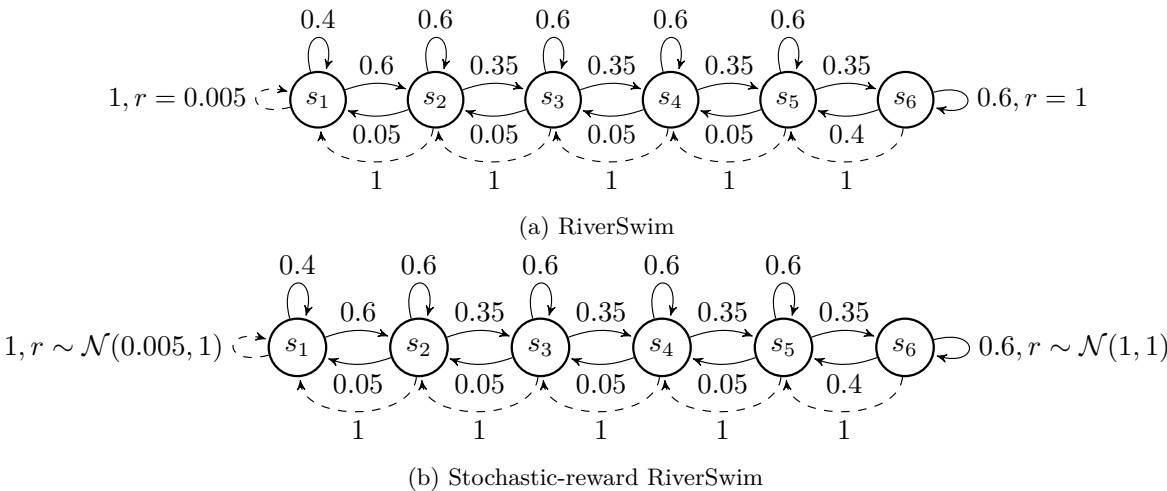

(a) RiverSwim

(b) Stochastic-reward RiverSwim

Figure 3: Diagram of RiverSwim and stochastic-reward RiverSwim. Dotted lines and solid lines show the transitions and probabilities for the left and right actions, respectively. Diagram adapted from Osband et al. (2013).

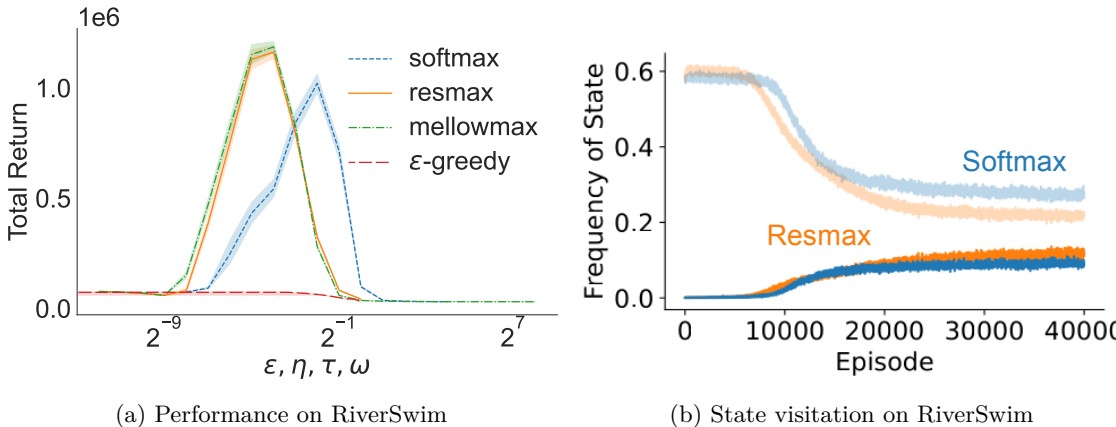

(a) Performance on RiverSwim

(b) State visitation on RiverSwim

Figure 4: (a) Performance across hyperparameters for RiverSwim. The means and bootstrapped 95% confidence intervals are shown for 30 runs for each parameter setting. The x-axis is plotted with a log scale, and shows the value of exploration parameters. We show results for $\varepsilon \in \{0, 0.1, ..., 1\}$ $\tau \in \{2^{-9}, 2^{-8}, ..., 2^2\}$, $\eta \in \{2^{-12}, 2^{-11}, ..., 2^0\}$ and $\omega \in \{2^{-12}, 2^{-11}, ..., 2^7\}$ (b) The state visitation frequency per episode for the initial state (light curve) and final state (dark curve) of RiverSwim for best hyperparameters ($\eta = 2^{-4}, \tau = 2^{-2}$). Resmax starts with slightly higher frequency in the initial low reward state, but once resmax encounters the final high reward state it pushes towards this state more than softmax since it avoids overemphasis of the low reward actions in the initial state. Note that mellowmax and epsilon-greedy are left out to mainly focus on comparing state visitation of resmax to softmax. But, as expected from the similar performance of mellowmax to resmax in these experiments, their state visitation is similar.

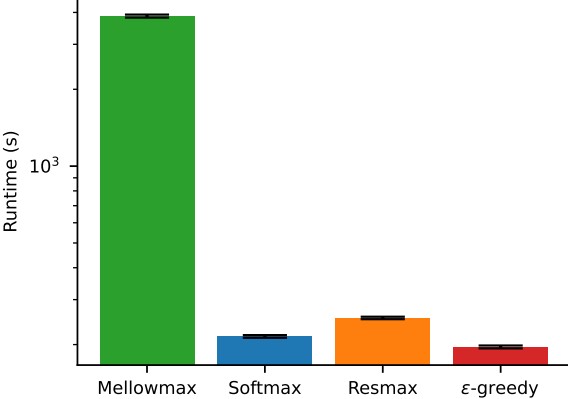

Figure 5: Logarithmic-scale runtime of Expected-Sarsa for each operator, averaged over all parameters across both RiverSwim variants. We show bootstrapped 95% confidence intervals with error bars.

We ran resmax, softmax, mellowmax and $\varepsilon$-greedy coupled with Expected Sarsa on RiverSwim across a broad range of hyperparameters ($\eta$, $\tau$, $\omega$ and $\varepsilon$). The hyperparameter ranges were chosen to match the technique. For example softmax requires relatively larger values of $\tau$ due its exponents. Results are shown in Figure 4a, where we measure algorithm performance by the total online return across these hyperparameters.

We find that resmax achieves a higher total return under best hyperparameter settings than softmax, indicating it is more capable of escaping the pull of the left state. $\varepsilon$-greedy fails to explore at all, and its total return is hardly visible for any values of $\varepsilon$. Figure 4b confirms that softmax lingers in the initial leftmost state longer than resmax does and ultimately spends less time in the high reward rightmost state. Mellowmax performs very similarly to resmax across hyperparamter settings; however, due to the call to a root finder within mellowmax, it takes much longer in terms of wall time (shown in Figure 5).

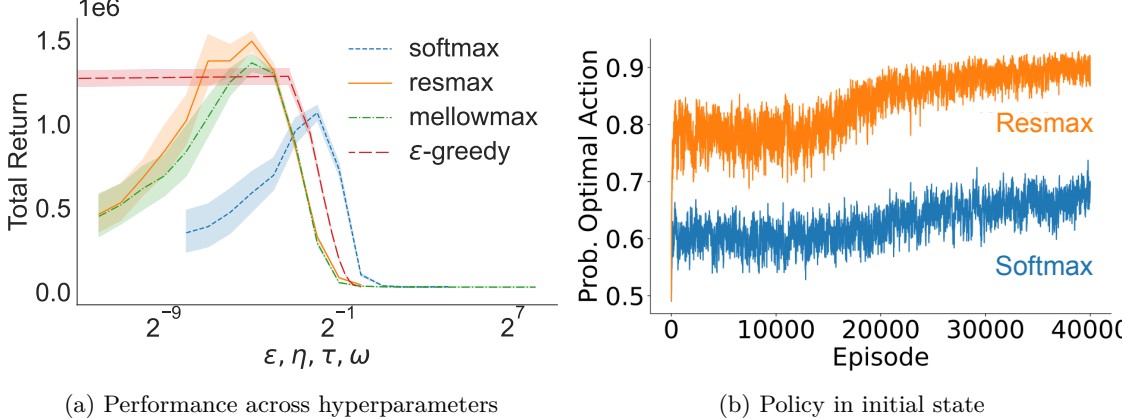

(a) Performance across hyperparameters

(b) Policy in initial state

Figure 6: (a) Performance across hyperparameters on Stochastic-reward RiverSwim. The means and bootstrapped 95% confidence intervals are shown for 30 runs for each parameter setting. The x-axis is plotted with a log scale, and shows the value of exploration parameters. We show results for the same hyperparameter ranges as in Figure 4a. (b) The probability of selecting the optimal action (right) from the initial state during the first step in the episode in Stochastic-reward RiverSwim. Results for both figures are averaged over 100 runs and are for optimal choices of hyperparameters in Stochastic-reward RiverSwim: $\eta = 2^{-5}, \tau = 2^{-2}$.

Stochasticity in rewards can help alleviate some of the misleading reward problem. Uncertainty in the rewards can prevent over-exploitation of sub-optimal actions and help drive exploration. We replaced the deterministic positive rewards in RiverSwim with rewards drawn from a Gaussian distribution with the mean being the original reward and variance of 1. Results are shown in Figure 6a. All four techniques see a reduction in hyperparameter sensitivity, but resmax, mellowmax and $\varepsilon$-greedy increase their total return under optimal parameter settings, whereas softmax does not. With stochastic rewards, softmax still assigns too great a weight on sub-optimal actions, as shown in Figure 6b. Softmax learns to take the optimal action in the initial state at a slower rate than resmax, indicating that softmax is concentrating too heavily on misleading rewards.

## 5 Resmax Is a Non-Expansion

The second key property of resmax is that it is a non-expansion. Figure 7a shows a simple two state MDP (taken from Asadi & Littman (2017)) where the action-values of softmax may not converge, as shown in Figure 7b. Resmax, since it is a non-expansion, converges for all settings of $\eta$. We show in Figure 7c that resmax indeed converges under a setting of $\eta$ that provides similar exploration to softmax.

We now show that the resmax operator is a non-expansion.

*Property* 5.1. The resmax operator is a non-expansion: for any two vectors $\vec{x}, \vec{y} \in \mathbb{R}^A$ and $\eta \geq 0$ we have,

$$|\mathrm{rm}(\vec{x}, \eta) - \mathrm{rm}(\vec{y}, \eta)| \leq \max_{i \in [A]} |x_i - y_i| \qquad (9)$$

*Proof.* Let us rewrite the resmax operator, using $\delta_i \doteq \|x\|_\infty - x_i$

$$\mathrm{rm}(\vec{x}, \eta) = \sum_{i \in [A]} \frac{x_i}{A + \frac{1}{\eta}\delta_i} + \|x\|_\infty \Big[1 - \sum_{i \in [A]} \frac{1}{A + \frac{1}{\eta}\delta_i}\Big]$$

$$= \|x\|_\infty - \sum_{i \in [A]} \frac{\delta_i}{A + \frac{1}{\eta}\delta_i} = \|x\|_\infty - \sum_{i \in [A]} \frac{\eta\delta_i}{A\eta + \delta_i} \qquad (10)$$

To show that resmax is a non-expansion, we use Theorem 1 of Paulavičius & Žilinskas (2006) which states the following. For Lipschitz function $f(x)$, $f : \mathbb{R}^d \to \mathbb{R}$,

$$|f(x) - f(y)| \leq L_1\|x - y\|_\infty$$

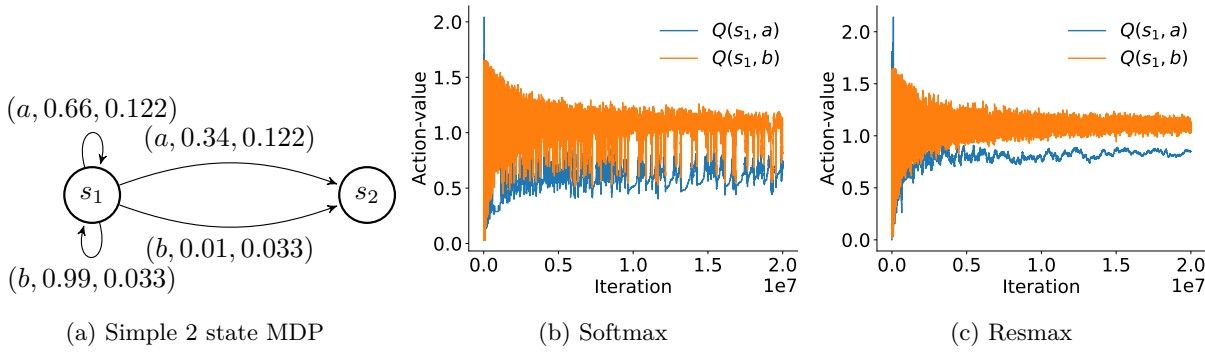

(a) Simple 2 state MDP                (b) Softmax                (c) Resmax

Figure 7: The non-expansion property is important for convergence under GVI. On the MDP shown in (a), if we use softmax in the Expected Sarsa update (with $\tau = 1/16.55$), the action values do not converge, as shown in (b). We can see that resmax does converge in (c) with $\eta = 0.000085$ (chosen to induce a similar level of exploration to softmax). We use a step size $\alpha$ at time $t$ to be $1/(\lfloor t/100,000 \rfloor + 1)$, which meets the usual conditions for stochastic approximation to converge (Sutton & Barto, 2018). The tuples in (a) are of the form: (action, probability of transition given action, reward). Results are smoothed with a window size of 10, as in Asadi & Littman (2017).

where $L_1 = \sup\{\|\nabla f(x)\|_1 : x \in D\}$ is the Lipschitz constant, $D$ is a compact set, and $\nabla f(x) = (\partial f/\partial x_1, ..., \partial f/\partial x_d)$ is the gradient of the function $f(x)$. Therefore, we need to show that the $\ell_1$ norm of the gradient of resmax is less than or equal to 1, namely $L_1 \leq 1$.

The resmax operator is differentiable, so we can apply this theorem. We characterize the partial derivatives and show that their $\ell_1$ norm is bounded by 1. Notice first that $\frac{\partial}{\partial x_j}\|x\|_\infty = 0$ if $x_j < \|x\|_\infty$ or if more than one entry in $x$ is maximal. Infinitesimally changing $x_j$ in these two cases does not change $\|x\|_\infty$. This partial derivative is only non-zero if $x_j$ is the unique max; changing it changes the max norm linearly, meaning the partial derivative is 1. Therefore, for $x$ where $x_j$ is the unique max,

$$\frac{\partial}{\partial x_j}\mathrm{rm}(\vec{x}, \eta) = 1 - \sum_{i \in [A], i \neq j} \frac{A\eta^2}{(A\eta + \delta_i)^2} \tag{11}$$

and for other indices (or $x$ where there are two or more maximal elements), we have

$$\frac{\partial}{\partial x_j}\mathrm{rm}(\vec{x}, \eta) = \frac{A\eta^2}{(A\eta + \delta_j)^2} \tag{12}$$

Notice now that

$$\frac{A\eta^2}{(A\eta + \delta_i)^2} = \frac{A\eta^2}{A^2\eta^2 + 2A\eta\delta_i + \delta_i^2} \leq \frac{A\eta^2}{A^2\eta^2} = \frac{1}{A}. \tag{13}$$

where the inequality holds because all the terms in the denominator are positive. The $\ell_1$ norm corresponds to summing up all the partial derivatives.

**Case 1** ($x$ with two or more maximal elements): All partial derivatives are of the form in Equation 12, bounded above by $1/A$ as per Equation 13, so the $\ell_1$ norm is bounded above by $\sum_{i \in [A]} 1/A = 1$.

**Case 2** ($x$ has a unique maximal element $x_k$): By Equation 13, we know that the partial derivative for $k$ (in Equation 11) is positive. Therefore, the $\ell_1$ norm corresponds to summing up all these nonnegative partial derivatives

$$\sum_{i \in [A], i \neq k} \frac{A\eta^2}{(A\eta + \delta_i)^2} + 1 - \sum_{i \in [A], i \neq k} \frac{A\eta^2}{(A\eta + \delta_i)^2} = 1$$

All these arguments were true for any $\vec{x}$, therefore, $L_1 = 1$, and so resmax is a nonexpansion. $\qquad\square$

This result also lets us show that the sequence of policies generated by approximate policy iteration under the resmax operator converges to a unique limiting policy regardless of the choice of the initial policy $\pi_0$. This follows from Theorem 1 of Perkins & Precup (2002), which simply requires the operator be Lipschitz continuous with an $\epsilon$-soft policy ($\pi$ is $\epsilon$-soft if $\pi(a|s) > \epsilon$ for $\forall a \in \mathcal{A}$). As shown in Section 4.1, the resmax policy is $\epsilon$-soft. By the non-expansion property of resmax, we can also say the resmax operator is Lipschitz continuous (Asadi & Littman, 2017).

A natural question is if we could have obtained a non-expansion result with a different variant of resmax. For example, one alternative is to take the squared max action gap, or more generally a p-norm: $(x_j - x_i)^p$. Somewhat surprisingly, we cannot do so; we prove that for $p > 1$, we would no longer have a non-expansion (see Appendix B.1). This further motivates the particular form we chose for resmax.

Finally, our strategy to prove that resmax is an non-expansion was simple: bound the $\ell_1$-norm of the gradient, with respect to the inputted action-values. A more complex, specialized approach was used for mellowmax (Asadi & Littman, 2017). Though our strategy is straightforward, it is the first time it has been used for these operators, and should facilitate analyzing other new operators. We use this strategy to show another potential operator, based on the escort transform (Mei et al., 2020), does not have the non-expansion property (see Appendix B.2).

## 6  Resmax in Deep RL

Many valued-based RL algorithms with function approximation use $\varepsilon$-greedy as a default method to add a degree of exploration. Resmax, in order to be considered a general-purpose method, should also scale to this setting. To show this, we conduct experiments in the deep RL setting (1890 experiments) across both easy and hard exploration Atari 2600 environments (Bellemare et al., 2013). Ranges for exploration parameters are chosen to ensure that the peaks for each approach resides in the chosen range. Further details on our implementation and computational infrastructure are in Appendix C.

We first study the utility of resmax compared to the baselines of mellowmax, softmax and $\varepsilon$-greedy on easy exploration Atari environments, namely Asterix and Breakout, chosen from human-optimal easy exploration environments as categorized in Bellemare et al. (2016). We begin by analyzing the sensitivity of both algorithms to their exploration parameters. The sensitivity plots for these experiments are shown in Figure 8. We see that softmax performs better with higher exploration parameters compared to resmax. This observation aligns with our intuition, considering the overemphasis property of softmax. In both environments, we can observe that the $\varepsilon$-greedy approach yields better results with a lower level of exploration, specifically with an

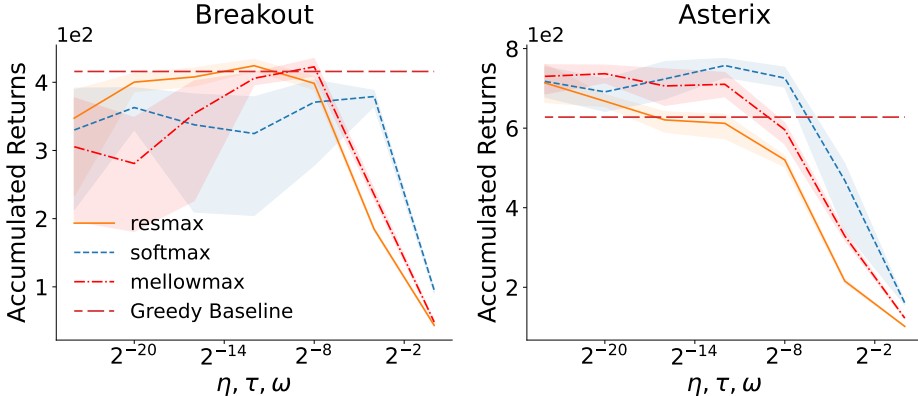

Figure 8: Sensitivity curves showing the performance across hyperparameters. Results shown are averaged over 10 runs for each parameter setting and the shaded region represents the standard error. The x-axis of sensitivity plots is plotted with a log scale, and shows the value of $\eta$ and $\tau$, which are swept over $\{2^0, 2^{-4}, \ldots, 2^{-20}, 2^{-24}\}$. $\varepsilon$-greedy is the best-performing instance selected from $\varepsilon$ values of $\{0.01, 0.1, 0.2, 0.3, 0.4, 0.5\}$.

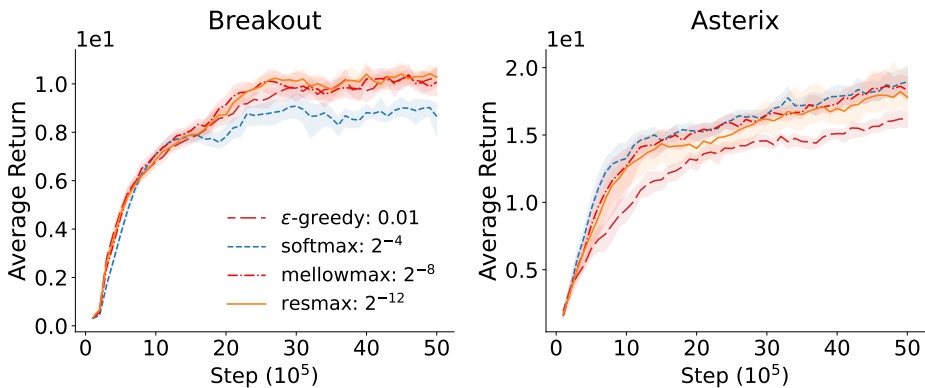

Figure 9: Learning-curves presenting the best performance across the selected exploration parameters. Results shown are averaged over 10 runs for each parameter setting and the shaded region represents the standard error.

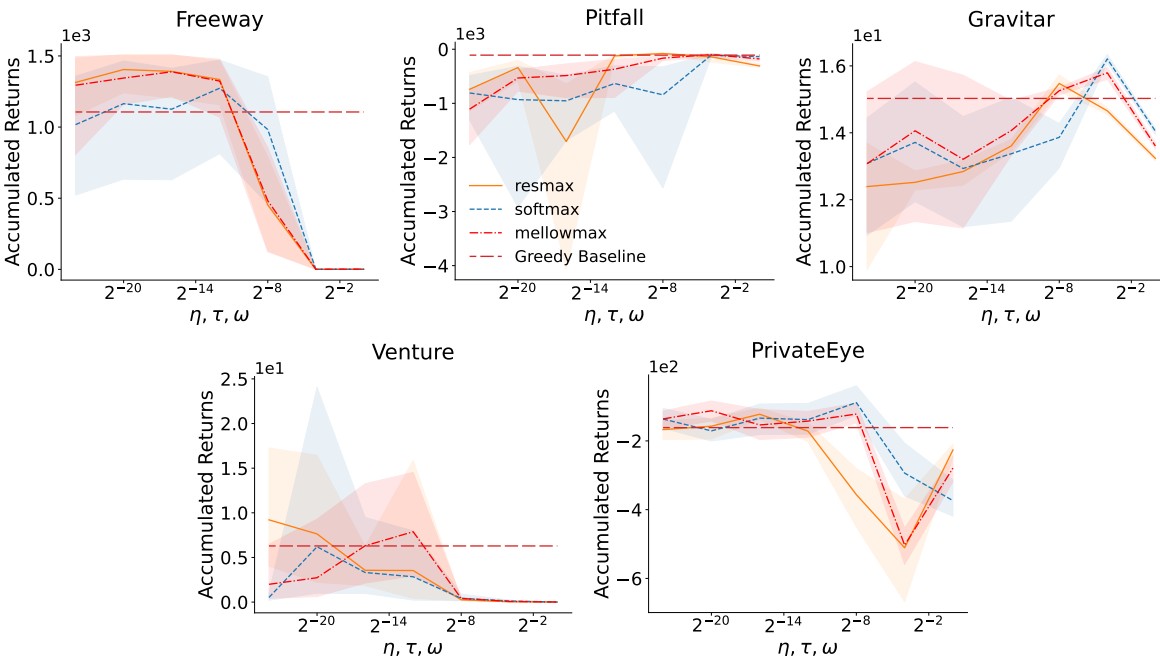

Figure 10: Sensitivity curves showing the performance across hyperparameters. Results shown are averaged over 10 runs for each parameter setting and the shaded region represents the standard error. The x-axis of sensitivity plots is plotted with a log scale, and shows the value of $\eta$ and $\tau$, which are swept over $\{2^0, 2^{-4}, \dots, 2^{-20}, 2^{-24}\}$. $\varepsilon$-greedy is the best performing instance selected from $\varepsilon$ values of $\{0.01, 0.1, 0.2, 0.3, 0.4, 0.5\}$.

$\varepsilon$ value of 0.01, indicating that these environments embrace more greedy algorithms, a finding consistent with Laidlaw et al. (2023).

The learning curves for the top-performing hyperparameter configurations of each algorithm are illustrated in Figure 9. In general, resmax exhibits superior performance or remains competitive when compared to softmax and $\varepsilon$-greedy, while demonstrating similar performance to mellowmax. Specifically, in the Asterix environment, resmax competes well with softmax and mellowmax, surpassing $\varepsilon$-greedy. However, on Breakout, resmax outperforms softmax, while performing similarly to $\varepsilon$-greedy and mellowmax.

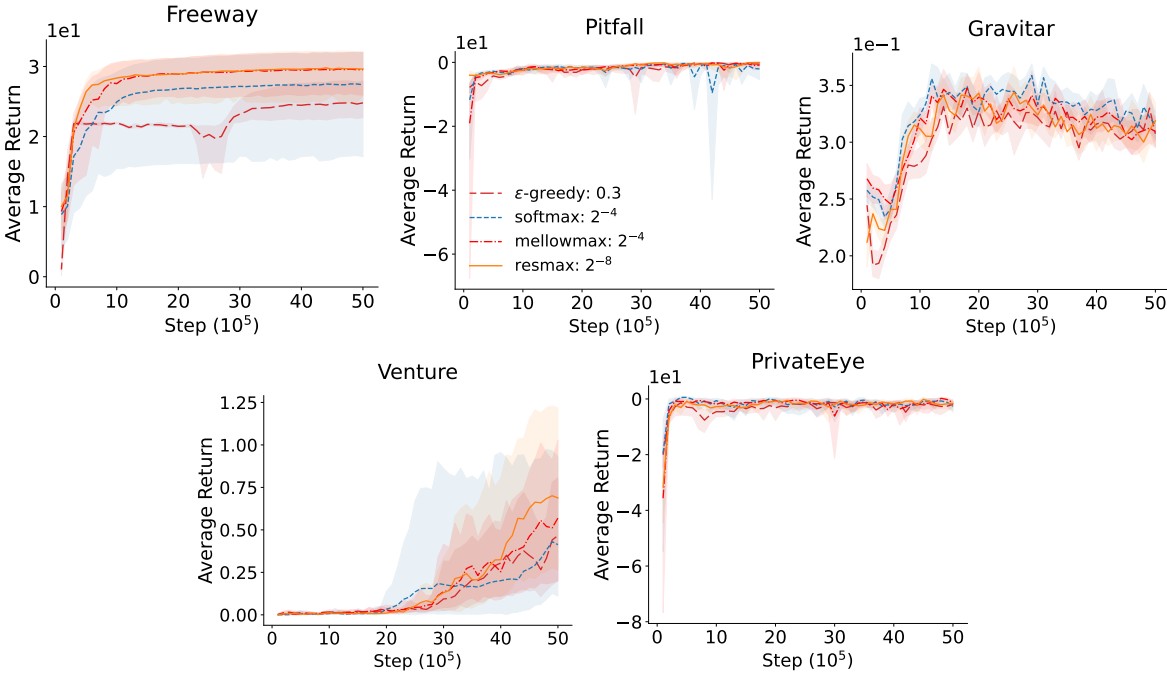

Figure 11: Learning-curves presenting the best performance across the selected exploration parameters. The results shown are averaged over 10 runs for each parameter setting and the shaded region represents the standard error.

Next, we analyze the efficacy of resmax with respect to the same baselines in five sparse-reward Atari environments that are hard to explore: Freeway, Pitfall, Gravitar, Venture, and PrivateEye. The sensitivity plots are presented in Figure 10. Opposite to the easy exploration results, resmax, mellowmax, and softmax tend to do better with lower exploration parameters, except in Gravitar and Pitfall, showing that they require more exploration in these environments.

The learning curves for the top-performing instances in these environments are depicted in Figure 11. As observed, resmax outperforms both softmax and $\varepsilon$-greedy in two of the environments: Freeway and Venture. Additionally, it exhibits competitive performance in the remaining three environments, with a slightly lower return in Gravitar compared to softmax. When compared to mellowmax, resmax performs similarly in all environments, except in Venture where it slightly outperforms mellowmax.

These findings indicate that resmax is a promising alternative to $\varepsilon$-greedy and softmax for promoting exploration. A particularly striking result is how much more effective both resmax and softmax are than $\varepsilon$-greedy, both in Atari and in earlier results. If nothing else, these results suggest that we should consider resmax and softmax more often in Atari experiments. Further, resmax is typically quite similar to softmax, with a few instances where it is notably better.

## 7   Conclusion

Soft-greedy operators, including $\varepsilon$-greedy, softmax and resmax, continue to play an important role in reinforcement learning. They serve dual purposes: they function as soft-greedification within the Bellman update and to induce a basic level of exploration. Hence, they are valuable for both on-policy and off-policy learning. Their simplicity is also a strength: they can be easily used with (deep) function approximation and are complementary to more focused exploration techniques. We propose a new soft-greedy operator, called resmax. Unlike softmax, resmax is a non-expansion, regardless of the choice of exploration parameter, and thus suitable for use with a Bellman update. Moreover, resmax ensures state-action space coverage and it avoids softmax's fundamental issue of overemphasis. Our empirical results show that resmax encourages more

exploration than softmax, since it does not overemphasize and also explores more efficiently than $\varepsilon$-greedy. Resmax also has the non-expansion property of mellowmax with a fraction of the computation required.

This paper proposes resmax in its simplest form, so there are many avenues for future research. As resmax is a new operator to reinforcement learning, it deserves further benchmarking and experimentation in both simple and complex environments in order to shed more light on when this operator is useful. Another natural direction for future work is to explore adaptation and normalization techniques which are suitable for resmax. Just like decay schedules for $\varepsilon$, schedules or adaption schemes for resmax could allow more exploration in early layer, and allow greedier policies later.

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

## A  Resmax Does Maximization

It can be easily shown that the Expected Sarsa update of resmax, represented in Equation 8, can do maximization (i.e., give the highest probability to the greedy action) when $\eta$ goes to 0. We first show how $\pi_{\mathrm{rm}}(a \mid s)$ for non-greedy actions will change when this happens:

$$\lim_{\eta \to 0} \pi_{\mathrm{rm}}(a \mid s) = \lim_{\eta \to 0} \frac{1}{|\mathcal{A}| + \eta^{-1}(q(s,b) - q(s,a))} = 0$$

This equality will hold as long as $q(s,b) \neq q(s,a)$.

Considering this, we can derive Equation 8 when $\eta$ goes to infinity as follow:

$$\lim_{\eta \to 0} \mathrm{rm}(q(s,\cdot),\eta) = \lim_{\eta \to 0} \sum_{a \notin G(s)} \pi_{\mathrm{rm}}(a \mid s) q(s,a)$$
$$+ (1 - \sum_{a \notin G(s)} \pi_{\mathrm{rm}}(a \mid s)) q_{\max}$$
$$= 0 - (1 - 0) q_{\max} = q_{\max}$$

Since $q_{\max}$ is the action-value of the greedy action, resmax can do maximization. It is also interesting to note that when $\eta$ goes to $\infty$, the generated policy will be equiprobable thus Expected Sarsa update of resmax will average all the action-values. So, this operator can make a balance between q-learning update and update with equiprobable policy by tuning the value of $\eta$, like softmax and mellowmax.

## B  More Non-Expansion Results

Here, we first show that a more general form of resmax operator is not a non-expansion, except when its form is equal to resmax. Then, we show that the escort transform operator recently presented in the policy gradient literature is not a non-expansion.

### B.1  Is the General Form of Resmax a Non-Expansion?

A general form of resmax operator with max action gap, $v$, replaced by a gap emphasis function $g(v) = v^p$ for $p \in \mathbb{R}^+$ is defined as follows

$$\mathrm{rm}(\vec{x},\eta) = \sum_{i \in [d], i \notin G(s)} \frac{x_i}{d + \frac{1}{\eta}(x_j - x_i)^p} + x_j \left[ 1 - \sum_{i \in [d], i \notin G(s)} \frac{1}{d + \frac{1}{\eta}(x_j - x_i)^p} \right] = \sum_{i \in [d], i \notin G(s)} \frac{x_i - x_j}{d + \frac{1}{\eta}(x_j - x_i)^p} + x_j \tag{14}$$

where $G(s) \doteq \{i \mid \max_i x_i = x_i\}$ and $x_j \doteq \max_i x_i$.

Now we want to show that if $p > 1$ resmax with gap emphasis function $g(v) = v^p$ is no longer a non-expansion. Let $d = 2, \eta = 1, j = 2$, and $\delta_1 \doteq x_2 - x_1 \geq 0$. We take the derivative of resmax with respect to $x_1$ and get

$$\frac{\partial}{\partial x_1} \mathrm{rm}(\vec{x}, \eta = 1) = \frac{-p\delta_1^p + \delta_1^p + 2}{(2 + \delta_1^p)^2} = \frac{2 + (1-p)\delta_1^p}{(2 + \delta_1)^2}.$$

Now we compute $\partial \mathrm{rm}(\vec{x}, \eta = 1)/\partial x_2$,

$$\frac{\partial}{\partial x_2} \mathrm{rm}(\vec{x}, \eta = 1) = \frac{(p-1)\delta_1^p - 2}{(2 + \delta_1^p)^2} + 1 = \frac{(p+3)\delta_1^p + \delta_1^{2p} + 2}{(2 + \delta_1^p)^2}.$$

Note that $\partial \mathrm{rm}(\vec{x}, \eta = 1)/\partial x_2 \geq 0$ when $\delta_1 \geq 0$ and $p > -3$. We will use this fact later. Since we have $\nabla \mathrm{rm}(\vec{x}, \eta = 1)$, we will define and compute $L_1(x)$,

$$L_1(\delta_1) \doteq \left| \frac{\partial}{\partial x_1} \mathrm{rm}(\vec{x}, \eta = 1) \right| + \left| \frac{\partial}{\partial x_2} \mathrm{rm}(\vec{x}, \eta = 1) \right| = \left| \frac{2 + (1-p)\delta_1^p}{(2 + \delta_1^p)^2} \right| + \left| \frac{(p-1)\delta_1^p - 2}{(2 + \delta_1^p)^2} + 1 \right|.$$

We would like to note that if $p = 1$ then $L_1(\delta_1) = 1$ for all $\delta_1 \geq 0$. Now we will show that if $p > 1$ then $L_1(\delta_1) > 1$ for some $\delta_1$. Since all the terms of $\frac{\partial}{\partial x_2}\text{rm}(\vec{x}, \eta = 1)$ are positive, we can remove the absolute value from this term. Now let $\delta_1 = (2c/(p-1))^{1/p}$ for $c \in [1, \infty)$, which when $p > 1$ is positive, then we have

$$L_1(\delta_1) = \left| \frac{2 + (1-p)(2c/(p-1))}{(2 + 2c/(p-1))^2} \right| + \frac{(p-1)(2c/(p-1)) - 2}{(2 + 2c/(p-1))^2} + 1$$

$$= \frac{(p-1)(2c/(p-1)) - 2}{(2 + 2c/(p-1))^2} + \frac{(p-1)(2c/(p-1)) - 2}{(2 + 2c/(p-1))^2} + 1 = \frac{4c - 4}{(2 + 2c/(p-1))^2} + 1.$$

Now for $c > 1$ we have that

$$L_1 \geq L_1(\delta_1) = \frac{4c - 4}{(2 + 2c/(p-1))^2} + 1 > 1$$

where the first inequality holds by the definitions of $L_1$ and $L_1(\delta_1)$ and the second strict inequality holds since $4c - 4 > 0$ when $c > 1$. Putting this all together, we have shown that if $p > 1$ for the gap emphasis function $g(v) = v^p$ and $\eta = 1$ then there exists a vector $\vec{x} \in \mathbb{R}^2$ such that if $\delta_1 \doteq x_2 - x_1 = (2c/(p-1))^{1/p}$ for $c \in (1, \infty)$, then $L_1 > 1$. This means that if we put more emphasis on the gaps, resmax is no longer a non-expansion.

This proof can be generalized for arbitrary $d$ by letting $j = \arg\max_i x_i = d$ and considering the special case when $\delta_1 = \delta_2 = ... = \delta_{d-1} \approx (cd/(p-1))^{1/p}$. Thus the current gap emphasis function for resmax is tight in the sense that adding more emphasis to the gaps would mean resmax is no longer a non-expansion.

### B.2 Is the Escort Transform a Non-Expansion?

In this section, we will show that a version of the escort transform does not have the non-expansion property. The escort transform (Mei et al., 2020) was introduced for policy gradient methods (with action preferences), but can naturally be defined for action-values as

$$\pi_e(a|s) = \frac{|q(s,a)|^p}{\sum_b |q(s,b)|^p}, \text{ for all } (s,a) \in \mathcal{S} \times \mathcal{A} \text{ and } p \geq 1.$$

Now we will analyze the following escort operator $\text{et}(q(s,\cdot), p) \doteq \sum_{a'} \pi_e(a'|s)q(s,a')$. Now let $\vec{x} \in \mathbb{R}^2$ be a two dimensional vector that lies in the first Cartesian quadrant, meaning the vector contains only non-negative elements. Then the escort operator becomes

$$\text{et}(\vec{x}, p) = \frac{x_1^{p+1} + x_2^{p+1}}{x_1^p + x_2^p}.$$

Note the absolute values are dropped because $\vec{x}$ is assumed to be in the first Cartesian quadrant. Now taking derivatives with respect to both $x_1$ and $x_2$ we get

$$\frac{\partial}{\partial x_1}\text{et}(\vec{x}, p) = \frac{x_1^{p-1}(x_1^{p+1} + (p+1)x_1 x_2^p - p x_2^{p+1})}{(x_1^p + x_2^p)^2}$$

and

$$\frac{\partial}{\partial x_2}\text{et}(\vec{x}, p) = \frac{x_2^{p-1}(x_2(x_1^p + x_2^p) - p x_1^p(x_1 - x_2))}{(x_1^p + x_2^p)}.$$

Now we want to compute the Lipschitz constant with respect to the max norm of $\text{et}(\vec{x}, p)$, $L_1 = |\frac{\partial}{\partial x_1}\text{et}(\vec{x}, p)| + |\frac{\partial}{\partial x_2}\text{et}(\vec{x}, p)|$. Let $\vec{x} = [1, 0]$ and $p = 1$, we have

$$L_1 = \left| \frac{1^{1-1}(1^{1+1} + (1+1)(1)(0)^1 - (1)(0)^{1+1})}{(1^1 + 0^1)^2} \right| + \left| \frac{0^{1-1}(0(1^1 + 0^1) - (1)(1)^1(1-0))}{(1^1 + 0^1)^2} \right| = \left| \frac{1}{1} \right| + \left| \frac{-1}{1} \right| > 1.$$

Thus we can conclude that the escort operator does not have the nlon-expansion property since we have an instance when $L_1 > 1$.

## C   Experimental Configuration

### C.1   Computational Infrastructure

We ran our experiments on a compute cluster. Each job used a single CPU core, except for Atari experiments that used GPUs. The compute cluster allocated CPUs based on availability. The possible options were 2.1Ghz Intel CPUs with model numbers E5-2683 V4 Broadwell, E7-4809 V4 Broadwell, or Platinum 8160F Skylake, as well 2.4Ghz Intel Platinum 8260 Cascade Lake. For GPU experiments, we used V100 Volta GPU. We also requested 400MB for the tabular setting. In the case of deep RL, we requested 16GB of memory for Atari environments. Different algorithms and exploration heuristics within one environment used the same configurations of resources.

### C.2   Logging Procedure

To save returns and steps per episode, we average returns or the number of steps per episode for all of the episodes that have been finished in a specific number of steps that we call log-interval. To elaborate, returns and the number of steps per episode for all the episodes that are finished in the log interval will be accumulated and averaged. We only store this final averaged value. For instance, if we set the total number of steps to $100,000$, and define a log-interval of $1,000$, then 100 values will be stored. This way of storing the results of our experiments can save us both memory and space. At the same time, the stored results are proportional to the performance of each of the employed algorithms. We use log-interval of $1,000$ for all our tabular experiments and log-interval of $100,000$ for all the Atari experiments.

### C.3   Hyperparameters

In this section, we present the hyperparameters that are used in our experiments and the reason for selecting them. One of the hyperparameters we needed to set fairly across different soft-greedy operators were their respective exploration parameters, such as $\eta$ and $\tau$. To do this, we swept over a large set of hyperparameters for each soft-greedy operator to make sure that the exploration parameter with the near-best performance resides in this set.

In the deep RL setting, we use the DQN algorithm. We chose a fixed set of parameters that work well across all three benchmark environments. These parameters are presented in Appendix Table 1. We swept over three different step sizes across all our experiments: 0.0005, 0.0001, 0.00005. Our experiments with these step sizes show that a step size of 0.0001 works best across all large-scale atari environments. We present the results in this paper based on these step sizes.

| Parameter Name | Fixed Value |
| --- | --- |
| Optimizer | Adam |
| $\beta_1$ | 0.9 |
| $\beta_2$ | 0.999 |
| $\epsilon$ for Adam | $10^{-8}$ |
| Batch size | 64 |
| Buffer size | $100,000$ |
| Number of training steps per iteration | 1 |
| Target network update frequency | $1,000$ |
| Number of steps before learning starts | $50,000$ |
| $\gamma$ | 0.99 |

Table 1: The fixed parameters used to run DQN experiments.

For implementing the neural networks we used the PyTorch framework. We used a convolutional neural network as the function approximation, with three convolutional layers that are followed by two fully connected layers. ReLU is used as the activation function for these networks. Convolutional layers have 32, 64, and 64 filters; a kernel size of 8, 4, and 3, and a stride of 4, 2, and 1, respectively. The first fully connected includes

512 neurons, and the second one outputs the action values. We use uniform Xavier initialization to initialize the weights of the network.

