# OpenReview forum: "Resmax: An Alternative Soft-Greedy Operator for Reinforcement Learning"
_TMLR — Accepted by TMLR_

### Review · Reviewer_Lagu · 2023-08-04

**Summary Of Contributions:**

The authors proposed a new soft-greedy operator for exploration in RL called Resmax, this operator uses the action-value gap while also avoids the overemphasize issue stemming from the exponential softmax operator. The authors furthered showed that their operator is a non-expansion. A series of toy examples are given to illustrate the intuition behind the new operator, the authors also demonstrated the effectiveness of their approach in DRL settings.

**Audience:**

Yes

**Claims And Evidence:**

Yes

**Requested Changes:**

One change I would recommend is for the authors to bring forward the definition of a non-expansion to the background section of the paper since it is a fairly important property of the ResMax operator and in the current paper does not actually come up until nearly towards the end.

**Strengths And Weaknesses:**

- Overall I like this paper a lot, the idea proposed by the authors is extremely straightforward but yet effective.
- The paper is very clearly written and nicely structured, the authors begin by enumerating the limitations of the most commonly used soft-greedy operators (mainly $\epsilon$-greedy and softmax, and to a lesser extent mellowmax) and very clearly shows how ResMax addresses each one of these limitations.
- Theoretical results in the paper are sound, the proofs are quite easy to read.
- On the theoretical results, I was also quite surprised that the general for of ResMax is not a non-expansion, do you have any intuition why this may be the case? Specifically why would placing more emphasize on the action gaps no longer make it a non-expansion?
- What I particularly like about this paper is how the authors use very simple and clear toy examples to demonstrate the properties of ResMax and they were very effective in making the results of the paper intuitive. I think this is a practice that should be promoted more within the RL community.
- One question I have for the authors is that it seems ResMax is limited to the setting of discrete action spaces, I was wondering if it is possible to extend the use of the operator to continuous action spaces. Since one advantage of a softmax is that it can be applied to both discrete and continuous action spaces

---

> ### Author Response · Authors · 2023-08-31
> **Author Response**
>
> We like to thank the reviewer for the detailed review. We discuss each of the points mentioned in the review below.
>
> **Understanding the Absence of Non-Expansion Property in the General Form of ResMax**
>
> It is true that the proof does not give this intuition. In general, we should not expect operators to be non-expansions, so the counter question is actually: why is resmax a non-expansion, intuitively? This is not that easy to answer, but here is some intuition (which we can add around the proof in the paper, and possibly flesh out a bit more with different examples). One setting where this might happen is that we have a large action gap, due to a large magnitude value, and the operator with p > 1 causes us to put even more weight on this large magnitude value, making the output bigger (we want our operator to keep things smaller if we want a contraction or non-expansion). Consider if we have two actions and eta = 1, with values q1 and q2, with q1 >> q2 (say q1 = 100, q2 = 1). Then the resmax operator has q1 (1-1/c) + q2 (1/c) for c = 2 + delta^p with delta = q2 - q1. If we use p = 2, instead of p = 1 that is used in resmax, the delta^p is much bigger and the weight 1/c on q2 is much smaller. Way more weight is put on the high magnitude q1 (overemphasis, like the softmax).
>
> **The Usage of Resmax on Continuous Action Spaces**
>
> This is a great point, and we have not yet considered how to extend this operator to continuous action spaces. It’s straightforward to use the same action gap, though we need to solve for a max over actions. We can remove the dependence on the size of the action set in the denominator, with a renormalization, though we may no longer have a minimal density for every action (nor is that possible for unbounded action spaces). Note that for continuous actions, too, we would likely not directly use the resmax policy; in SAC for example, the Boltzmann (softmax) policy is only used as a target in the KL. A parameterized policy is learned to avoid directly sampling from the softmax policy. This consideration would likely also play a factor, if we only needed to define a sensible resmax continuous action policy to use as a target in the KL. In general, there are non-trivial details to be worked out, and our focus in this work was only discrete actions. We could add a short discussion on avenues to obtain such an extension in the conclusion.
>
> **Bringing Forward the Definition of Non-expansion**
>
> That’s a great suggestion. We will introduce what it means to be a non-expansion earlier.

---

### Review · Reviewer_YcUX · 2023-08-08

**Summary Of Contributions:**

The authors introduce Resmax, a soft-greedy operator for value-based RL approaches that combines the advantages of the $\epsilon$-greedy (full coverage of action space) and softmax (sampling prioritizes promising actions) operators while avoiding their downsides (value agnosticism and over-concentration, respectively). It shares these properties with the Mellowmax operator while retaining the computational efficiency of standard operators. Unlike the softmax operator, it is a non-expansion. Experiments are carried out comparing the performance of these operators in both the tabular and function approximation settings.

**Audience:**

Yes

**Broader Impact Concerns:**

I do not have any broader impact concerns.

**Claims And Evidence:**

Yes

**Requested Changes:**

- I would appreciate  experimental results and/or discussion regarding the combination of Resmax with strong exploration methods. Are such approaches complementary?

- Any deep RL results beyond Atari would also be appreciated, though not necessary.

- I would also like to see more discussion regarding why Resmax does not provide real gains on deep RL tasks---is there some property of a task/agent that would make Resmax more useful?


**Strengths And Weaknesses:**

Strengths

- The paper is clearly written and structured. The presentation is straightforward.

- The proposed operator is original and its benefits are clearly articulated.

- The theoretical analysis supports the claimed properties of Resmax.

- The empirical evaluation is thorough.

Weaknesses

- The main weakness is that for the large-scale problems tested by the authors, there doesn’t seem to be any real performance benefit in using Resmax. I don’t see this method being widely adopted in practice without a convincing demonstration of its empirical effectiveness.  I would be most curious about Resmax’s interaction with state-of-the-art exploration methods that can solve challenging tasks like Montezuma’s Revenge (and the other exploration domains presented). I would also be curious to see performance on deep RL tasks beyond Atari.

- There is not much discussion which fosters understanding of Resmax's performance on the deep RL tasks.

---

> ### Author Response · Authors · 2023-08-31
> **Author Response**
>
> We appreciate your positive comments about the work, and understand the request for more insights into resmax. There is more to be done to truly understand this operator. It is rare that one paper provides the complete picture, and follow-up work (even work simply testing out the resmax operator in their setting) can help get a deeper understanding over time. This paper introduces this operator and provides a solid foundation for its properties. We agree more results beyond our current Atari results would help get a more thorough picture of how resmax performs in the wild, when combined with existing methods. But, we do see that as future work, and appreciate that you also acknowledge that it is a nice to have.
>
> Now let us address your two other questions.
>
> **Resmax is complementary to other exploration techniques**
>
> You are absolutely right that resmax is complementary to strong exploration approaches. Often a new exploration approach still uses a basic stochastic policy, such as epsilon-greedy or softmax (Boltzmann). Resmax can similarly be used in place of these. In this work, we tested resmax in the most typical algorithms (e.g., DQN), since it remains highly common to use algorithms like DQN without smarter exploration. Absolutely a next step for us is to layer resmax on algorithms that do directed exploration, such as those that include optimism in the action-value estimate. We can add a short discussion about this in the conclusion.
>
> **Why is resmax not better in Atari?**
>
> We first want to state that we do in fact see that resmax improves at least a little bit over softmax and epsilon-greedy. Now, it is hard to say why we do not see as big of a gap here, since there are more complexities in these experiments. In general, the purpose of these experiments was not to make very careful scientific claims about why one method might be better than another (that was the purpose of the earlier experiments). Rather it was to demonstrate the behavior of resmax in this more difficult setting, primarily as a sanity check: it is at least comparable to existing methods and was easy to implement.
>
> Nonetheless, let us speculate a bit about why we do not see as big of a gap here. One possibility is that we as yet do not have particularly good deep RL systems, and performance is dominated by this fact. Another possibility is that resmax could be more effective with a scheduling technique for its parameters, like the ones typically used for epsilon-greedy, to allow the policy to become more greedy. In the other experiments, we found bigger differences under non-stationarity and misleading rewards (one where overestimation causes issues). These Atari environments do not have these properties. Pinpointing the exact cause, however, when there are many other confounding factors, would be difficult. Rather, as mentioned above, the primary purpose of these results in Atari was to be demonstrative.

---

> > ### Comment · Reviewer_YcUX · 2023-09-18
> > **Response**
> >
> > Thank you to the authors for your response! I think these points are reasonable, and I stand by my overall positive assessment of the paper.

---

### Review · Reviewer_G7Vg · 2023-08-22

**Summary Of Contributions:**

The paper proposes Resmax, a control operator as an alternative to Softmax, Mellowmax and classic greedy operator. Resmax averages over Q-functions using a policy that is defined using the action gaps between actions. The paper shows that Resmax is a non-expansion, and entails better exploration property compared to Mellowmax. The paper demonstrates some empirical gains on tabular hard exploration problems and improvements in some deep RL settings.

**Audience:**

Yes

**Broader Impact Concerns:**

No.

**Claims And Evidence:**

No

**Requested Changes:**

I have some questions that I hope the authors can respond to.

=== **Eqn 4** ===

Is there a typo in Eqn 4? The Mellowmax operator should have $x_i$ in the exponents, while the current form in the paper only has $x_i$ inside the log.

=== **Non-expansion vs. contraction of the operator** ===

I think a major theoretical bottleneck here is that Mellowmax operator is contractive, regardless of the temperature parameter $w$, while Greedy operator as a limiting case is always contractive too. This implies that when iterating both operators on a set of Q-functions, we expect convergence. This might be a major motivation to replace Softmax by Mellowmax, since the former does not guarantee convergence as shown by Asadi et al.

Since Resmax is only a non-expansion, there is no formal guarantee that it converges during iterations. Will we be able to identify cases where it does not converge? Like the two fixed point of Softmax case identified in Asadi. I think this would be a valuable finding in general. Meanwhile, since Resmax is not proved strictly contractive or convergent, we may not be able to expect an improvement over alternative operators in terms of value based learning.

=== **Other properties of the operators** ===

I feel there can be many more theoretical properties of the operator worth discussing. What's the fixed point? Is the fixed point unique? At fixed point do we recover optimal policy or entropy regularized policy? These properties will distinguish Resmax from Softmax and greedy even in case the Resmax operator is proved convergent, and will be useful to show.

=== **Empirical performance Fig 2 & 4** ===

Minor: x-axis in Fig 2, do we mean all three parameters for different algorithms take the same value on the axis? Why not use the same temperature parameter to represent them all, as they represent similar quantities.

In Fig 4 on the right, why do we not have a comparison of the state visitation between Resmax and Mellowmax, are they very similar?

Fig 5, it'd be nice to have log plots for the right three algorithms, it is hard to tell the difference from the current plot.

=== **Deep RL** ===

Fig 9: it looks like Resmax does not achieve significantly better performance compared to Softmax nor Mellowmax? In certain cases, eps-greedy performs quite well (and in the reviewer's experience, eps-greedy is a very strong baseline in Atari domains). The testing environments are breakout and asterix, which may not be the best testing benchmark for hard explorations. There are a number of exploration oriented games in the Atari suite, which eps-greedy struggles to solve, why not benchmark on those envs since we are comparing the exploration ability?

Fig 10: do we plot the final performance of these algorithms on a suite of games? It is a bit hard to tell what conclusion to make of the plot, since it seems that Resmax is just as sensitive to other algorithms wrt the temperature parameter, it is hard to argue that Resmax offers empirical gains in terms of final performance nor robustness.

=== **Why Resmax** ===

A central question is: why would one tend to use Resmax compared to alternative operators? Resmax is not provably convergent compared to Mellowmax. And although by design Resmax enjoys certain intuitive empirical properties such as no-early-commitment compared to Softmax, there should be a spectrum of operators that enjoy similar properties. Then, why the specific form of Resmax operator? Empirically, it is also hard to see the performance gains achieved by Resmax compared to others, over standard benchmark tasks. I think the paper can be further improved by strengthening both theoretical and empirical results.

**Strengths And Weaknesses:**

The idea of alternative operator to Softmax, Mellowmax and greedy operator is interesting, since it can potentially point to promising new directions to learn Q-functions while ensuring better exploration. This paper should be of interest to general RL audience, including theorist and practitioners.

However, overall I feel the results presented in the paper are a bit underwhelming. The theoretical result is essentially the non-expansion property, while advantages on better exploration are more or less based on empirical arguments. The empirical gains are most presented on tabular domains, whereas the improvements on deep RL domains are not very statistically significant. This makes it difficult to consider the Resmax operator as a well-balanced alternative to other existing operators.

---

> ### Author Response · Authors · 2023-08-31
> **Author Response**
>
> Thank you for your in-depth review. Your comments will absolutely help us make the paper better.
>
> **Equation 4**
>
> Good catch. We will fix this in the final version of the paper.
>
> **Non-expansion vs. contraction of the operator**
>
> We believe there is a small confusion here. The non-expansion property is what is required to ensure convergence of policy iteration. Asadi and Littman showed that mellowmax is a non-expansion, and point out that the softmax is not a non-expansion. Note also that the max operator is also only a non-expansion.
>
> However, this comment is likely coming from the fact that the Bellman operator needs to be a contraction. This we get due to having a discount that is less than 1, or other conditions in the episodic setting with no discounting. For example, the non-expansion property plus a discount less than one gives a Bellman operator that is a contraction. We will highlight this in the paper, to avoid this confusion.
>
>
> **Other properties of the operators**
>
> We partially answered this question above: the standard policy iteration theory applies, and we know the update converges to a unique value function (by Littman and Szepesvari). We will also mention in the paper that this converges to a soft-optimal policy, because the resmax operator maintains a stochastic policy.
>
> **Figure 2**
>
> To be fair, we used the same number of hyperparameters for each operator. Each operator tends to perform well in different ranges in the tabular case. That was why we chose different ranges for each operator.
>
> **Figure 4**
>
> We left out mellowmax in this figure because we wanted to mainly focus on comparing resmax to softmax and epsilon-greedy. But, as expected from the similar performance of mellowmax to resmax on these experiments, their state visitation is similar. We will add this as a comment to the paper, so that the reader is not wondering why it is not included. The plot is provided in this link: https://anonymous.4open.science/r/temp_repo-8D66/riverswim_state_visit.pdf
>
> **Figure 5**
>
> Nice point about the log plots. We have made that here:
> https://anonymous.4open.science/r/temp_repo-8D66/runtime_log_scale.pdf
>
> We will replace our current plot in the main body with this log plot.
>
> **Deep RL**
>
> We agree that the differences on Atari are smaller. However, one of the primary aims here was not to show that resmax is state-of-the-art, but rather to demonstrate that it is feasible and appropriate to use this new operator in more difficult settings when layered on existing algorithms (one might say “in the wild”). We tested its properties more carefully in the small environments. Many confounding factors arise in Atari, and so again our primary purpose of these experiments is as a sanity check: resmax is at least comparable to existing methods and there were no barriers to implementing and using it in Atari (in fact, it was easy to implement).
>
> You make a reasonable point that there are some harder exploration environments that might show a bigger difference. We do think it is actually a nice result that in environments where epsilon-greedy is known to do well, resmax does as well. Further, if you are considering some extremely hard exploration problems like Montezuma's revenge, then our simple operator is unlikely to produce sufficient exploration (we discuss this a bit in the paper). The resmax policy provides some stochasticity, like epsilon-greedy and softmax, but on its own is not a direct exploration approach. In general, we did not carefully pick environments where we thought resmax might do the best, because our primary goal was to see: in standard environments with standard algorithms, how does resmax behave? It is a success when a new approach is as easy to use in existing systems, as the existing approaches for which we have tuned for much longer (like epsilon-greedy).
>
> **Why Resmax**
>
> We hope this concern is allayed by the clarification of resmax’s theoretical properties. We do agree that more needs to be done to understand when you might prefer resmax to these other options. This first paper introduces this operator and provides a solid foundation understanding its properties. It gives us another well-understood tool. We do hope others will test out this operator, and as a community we can better understand when it is preferred.

---

### Decision · Action_Editors · 2023-09-30

**Recommendation:** Accept with minor revision

**Comment:**

This paper proposes a new soft-greedy operator, called resmax, in which the probability of taking an action is (inversely) proportional to the gap between their (estimated) value and the one of the (estimated) optimal action. Resmax operator tries to combine the benefits of previous soft-greedy operators avoiding their main flaws, i.e., random dithering for $\epsilon$-greedy, over-concentration for softmax, and high computational cost for mellowmax. The resulting operator is simple, intuitive, and easy to plug in the existing value-based algorithms, while bringing good theoretical properties.

The reviewers found the resmax operator to be interesting and well-presented, providing a mostly positive evaluation of the paper. On the other hand, all the reviewers ultimately agreed that the experimental evaluation, especially in deep RL domains, does not fully support the empirical superiority of resmax against previous operators, which would make it hard to convince practitioner to employ resmax in place of more established operators, potentially limiting the impact of the work. I think this is a valid concern. However, as one reviewer rightly noted, the latter is "largely orthogonal to the acceptance criteria for TMLR".

Hence, I am providing an Accept recommendation while encouraging the authors to implement a few minor changes to the manuscript, which are listed below.

- Change the last sentence in the Introduction to avoid overstating the empirical superiority of resmax;
- Fix equation 4;
- Highlight why non-expansion of the operator is enough to ensure convergence of value-based methods;
- Explicitly comment on why mellowmax curve is not in Figure 4;
- Incorporate the new version of Figure 5 with log scale;
- Please use \citep when authors names are not part of the sentence;
- Background, second paragraph: to learns -> to learn;
- In the plots, consider using line patterns to ease the visual inspection of the curves and reporting bootstrapped confidence intervals instead of standard errors;
- Figure 5, 8, 9 can be reduced in size without hurting legibility.

**Audience:**

This paper can be of general interest to the community of value-based RL, in which soft-greedy operators are backbones of the most successful algorithms.

**Claims And Evidence:**

The reported claims are (mostly, see below) supported by either theoretical derivations, such as for the non-expansion property, or empirical evidence, such as showing that resmax is less prone to overcommitting than softmax and computationally less demanding than mellowmax.

As reviewers noted, the experiments do not fully support clear empirical benefits from the use of resmax in deep RL. Thus, I encourage the authors to rephrase the last sentence in the Introduction (before Remark paragraph) to avoid any overstatement.

---

> ### Author Response · Authors · 2023-11-02
> **Final Revision**
>
> Thank you for your helpful suggestions.
>
> We applied all of the suggestions to our paper except updating the plots with line patterns and bootstrapped confidence intervals. The reason is that a member of the team with the data is away until Dec. 1. This means we could not change the plots. If you think that is essential to publishing the paper, we are happy to update the paper when they get back.

---

> > ### Comment · Action_Editors · 2023-11-04
> > **Camera-ready version**
> >
> > Dear authors,
> >
> > I have gone through the new version and the requested changes are properly implemented to me.
> >
> > In my opinion, the plots are not looking great and I am not sure results would be fully legible in a black-and-white printed version. Anyway, TMLR does not enforce any strict guidelines on the style of the plots, so it is up to you whether you want to update the plots or keep them as is.
> >
> > Please submit a camera-ready version (compiled with \usepackage[accepted]{tmlr}, see the guidelines at https://github.com/JmlrOrg/tmlr-style-file/blob/main/main-accepted.pdf) whenever you want the final version to be published.
> >
> > Best,
> > Action Editor

---

> > > ### Comment · Action_Editors · 2023-12-12
> > > **Reminder**
> > >
> > > Dear authors,
> > >
> > > This is to remind you that you didn't submitted a final version of the paper yet.
> > >
> > > Note that the paper cannot be published until then.
> > >
> > > Best,
> > > Action Editor